# Flocks of Stochastic Parrots: Differentially Private Prompt Learning for Large Language Models

**Haonan Duan**[*][†]**, Adam Dziedzic**[†]**, Nicolas Papernot, Franziska Boenisch**
University of Toronto and Vector Institute

## Abstract

Large language models (LLMs) are excellent in-context learners. However, the sensitivity of data contained in prompts raises privacy concerns. Our work first shows that these concerns are valid: we instantiate a simple but highly effective membership inference attack against the data used to prompt LLMs. To address this vulnerability, one could forego prompting and resort to fine-tuning LLMs with known algorithms for private gradient descent. However, this comes at the expense of the practicality and efficiency offered by prompting. Therefore, we propose to privately learn to prompt. We first show that *soft* prompts can be obtained privately through gradient descent on downstream data. However, this is not the case for *discrete* prompts. Thus, we orchestrate a noisy vote among *an ensemble of LLMs* presented with different prompts, i.e., *a flock of stochastic parrots*. The vote privately transfers the flock's knowledge into a single public prompt. We show that LLMs prompted with our private algorithms closely match the non-private baselines. For example, using GPT3 as the base model, we achieve a downstream accuracy of $92.7\%$ on the sst2 dataset with $(\varepsilon = 0.147, \delta = 10^{-6})$-differential privacy vs. $95.2\%$ for the non-private baseline. Through our experiments, we also show that our prompt-based approach is easily deployed with existing commercial APIs.

## 1 Introduction

Large language models (LLMs) exhibit strong capabilities for in-context learning [6, 40]. By prepending the adequate prompt to an LLM's input, the model can perform a myriad of natural language downstream tasks without any modifications to its parameters [41]. While the data used to train an LLM is usually assumed to be public, downstream data used in the prompt is often more sensitive. This can elicit confidentiality issues, for instance, if prompts contain information that represents valuable intellectual property [34]. At the same time, it also raises privacy concerns when the data involves personal information about individuals.

In this paper, our first contribution is to show that these concerns are valid. We are the first to instantiate a highly effective membership inference attack (MIA) [7, 45] against prompts. Our attack is able to determine if a given data point was used within the prompt of the LLM. The only existing solution to mitigate this privacy risk would be to forego prompting and instead fine-tune the LLM with a privacy-preserving training algorithm [25, 54]. Yet, fine-tuning lacks the efficiency and practicality of prompting. Indeed, fine-tuning requires significantly more data [42], computational resources [25], and storage space [26]. Additionally, fine-tuning requires access to the LLM parameters. However, many of the state-of-the-art LLMs are proprietary models deployed behind an API which only allows its users to query the LLMs [3, 6, 10, 17, 35].

---

[*]Corresponding and leading author: haonand@cs.toronto.edu
[†]Equal contribution.

37th Conference on Neural Information Processing Systems (NeurIPS 2023).

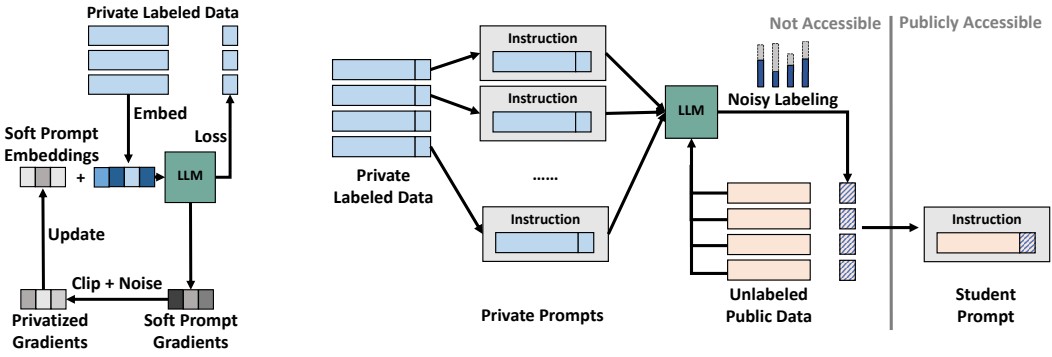

(a) PromptDPSGD                                  (b) PromptPATE

Figure 1: **Our methods for private prompt learning.** *Left:* PromptDPSGD obtains the input gradients from the LLM, and performs DPSGD to update the soft prompt embedding while keeping the LLM frozen. *Right:* PromptPATE creates a noisy ensemble of private discrete prompts, and then transfers knowledge by selecting a student prompt that can be publicly released. PromptPATE only needs black-box access of the LLM and, thus, can be easily deployed with commercial APIs.

To leverage the benefits of prompting while at the same time protecting the data contained in prompts, we propose the first algorithms for prompt learning with privacy. Our algorithms offer rigorous guarantees expressed using differential privacy [15]. Perhaps closest to existing work on fine-tuning, we propose to leverage the canonical DPSGD algorithm [1] to learn soft promptswith differential privacy guarantees. Our PromptDPSGD algorithm performs a private gradient descent on the soft prompt embeddings prepended to the LLM's private input. Since these embeddings have very few parameters in comparison to LLMs, our PromptDPSGD is efficient and yields competitive privacy utility trade-offs at a fraction of the training complexity of private fine-tuning.

However, learning soft prompts with DPSGD may not always be possible because it requires computing gradients with respect to the prompt input. As mentioned previously, current APIs [3, 6, 10, 17, 35] usually do not provide these gradients. We thus turn to *discrete* prompts which consist of natural language tokens. Discrete prompts address the aforementioned limitations while being more data-efficient. Our insight is to observe that LLMs with discrete prompts naturally lend themselves to another canonical approach of differentially private learning known as the private aggregation of teacher ensembles (PATE) [37]. We introduce PromptPATE, which creates an ensemble of LLMs with different discrete prompts from the private dataset which we refer to as *a flock of stochastic parrots* [5]. Since interacting with the flock directly can leak private information about the prompts, as we demonstrate with our MIA, PromptPATE additionally performs a knowledge transfer. Therefore, each model in the flock generates a next token prediction for a short input sequence of some public data. By performing a noisy majority vote over all models' token output, we generate a single output that, due to the noise addition, implements differential privacy guarantees while incorporating knowledge from the flock. The public input together with the noisy aggregated output form a new single example for the discrete *student prompt* that can be prepended to the LLM in lieu of the individual prompts which contain private information. In addition to providing rigorous privacy guarantees, our PromptPATE is highly efficient, since, instead of having to query every model from the flock at inference time, it suffices to query the LLM prepended with the student prompt *once*.

We perform extensive experiments against multiple popular LLMs, such as GPT3 [6] and Claude [3], that are deployed behind commercial black-box APIs. Our results highlight that PromptPATE provides high downstream performance that matches the one of non-private prompting even at very strong privacy guarantees. On the sst2 dataset with GPT3, for instance, we reach an accuracy of 92.7% with privacy costs as little as $(\varepsilon = 0.147, \delta = 10^{-6})$-differential privacy, even when the public data used during PromptPATE's knowledge transfer stem from a different distribution than sst2. Our results closely matches the non-private baseline accuracy (95.2%). Thus, we conclude that prompt learning for LLMs is not only more efficient and practical than fine-tuning but can also achieve high utility even with strong and practical privacy protection in place.

In summary, we make the following contributions:

- We instantiate the first MIA on prompted LLMs and show that we can effectively infer membership of the prompted data points with high success.

- We propose a lightweight alternative to DP fine-tuning, namely PromptDPSGD, which optimizes orders of magnitude fewer parameters while keeping the original LLM frozen.

- We propose PromptPATE, the first method for DP learning with LLMs that requires only black-box access to the model—making it easily deployable for commercial LLM APIs.

- Our experiments on multiple state-of-the-art commercial APIs [6, 3] highlight that our methods achieve both high utility and strong privacy protections in various setups.

## 2 Background and Related Work

**Prompts for LLMs.** The success of LLMs, such as BERT, Claude, OPT, or different versions of GPT and their exceptional in-context learning capacities gave rise to prompt-based learning [14, 6, 39, 40, 35, 56]. Prompts serve as *demonstrations* of the downstream task, which the model can then generalize from. There are two paradigms for LLM prompting, namely *discrete* and *soft* prompts.

Discrete prompts [6, 16, 18, 27, 44] are natural-language instructions that contain examples from the downstream task in a well-crafted template. Tuning discrete prompts is often done by prompting the model with different combination of examples, assessing their performance on the downstream task, and choosing the combination that yields the highest performance as the final prompt.

In contrast to discrete prompts, soft prompts [24, 27] prepend trainable continuous embeddings to the inputs of LLMs. These embeddings are initialized either at random or with embedding vectors that correspond to tokens from the dictionary. During tuning, the embeddings are updated through gradient descent to minimize the loss of the prompted model on the private downstream task. To increase performance further, trainable embeddings can be prepended not only to the input but also to every LLM layer, a technique known as *prefix* [26, 28, 29].

Both soft prompts and prefix train end-to-end without any human involvement through backpropagation over the LLM. On the other hand, discrete prompts have to be designed manually through careful prompt engineering. Yet, prompt engineering only needs inference passes over the LLM which makes discrete prompt more computationally lightweight. Our work provides privacy protection for all of these paradigms: discrete prompts, as well as for soft prompts, and prefix.

**Privacy Leakage in LLMs.** LLMs have been shown to memorize data both from their original large training corpora [8, 20, 23, 32, 48, 55] and from smaller private datasets used to fine-tune them for downstream tasks [33]. The only prior work around privacy leakage in prompt-based learning utilizes prompts to extract knowledge from trained LLMs [13, 22, 38]. In contrast, we study the privacy of the prompting data itself. To do so, we investigate the canonical privacy attack known as **membership inference attacks (MIA)** [7, 45]. Its use as a practical means to demonstrate leakage of private information in ML was recently popularized by a line of work on quantifying memorization [9, 43, 47]. While prior work utilizes MIAs to assess whether a given data point was used to train an LLM, we instantiate a MIA to assess whether a given data point was used within the prompt prepended to the inputs of a trained LLM.

**Defending Against Privacy Leakage in LLMs.** Prior work either focuses on training [2, 19] or fine-tuning [25, 54] LLMs with privacy guarantees. These approaches rely on the mathematical framework of **differential privacy (DP)** [15] and in particular the **DPSGD** algorithm for private stochastic gradient descent [1]. Here, DPSGD is applied to guarantee that one outputs approximately the same model parameters whether or not any given data point was used to train or fine-tune the model. To achieve this, DPSGD clips the per-example gradients that are computed during training and adds well-calibrated noise to each model update. These two operations typically increase the computational complexity of training and decrease the utility of the resulting model [1, 4, 49]. To counteract these effects, state-of-the-art methods for full DP-fine tuning in LLMs require extensive hyperparameter tuning and vast computational resources [25]. Alternative approaches refrain from updating the large number of model parameters and instead introduce additional layers into the model architecture and only fine-tune these layers with DPSGD [54]. To the best of our knowledge, no prior work attempted to provide DP guarantees for prompt data in LLMs.

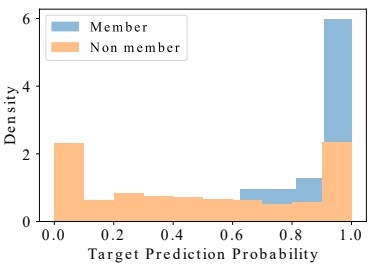

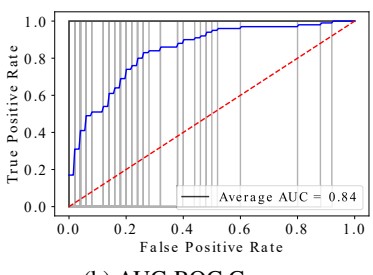

| (a) Probability Distribution. | (b) AUC-ROC Curve. |

Figure 2: **MIA Risk.** We study GPT3 prompted with 100 different one-shot examples (dbpedia). *left*: We present the prediction probabilities at the correct class for members (the one-shot example) and non-members (50 randomly sampled private points). The output probability for members is significantly higher than for non-member data points. *right*: We present the AUC-ROC curves of our MIA against the 100 prompts (gray lines) and the blue line as an average over all attacks. Given that each prompt has only one member, the resulting TPRs can only be 0% or 100% which leads to the step-shape of the gray curves. The result indicates that our attack is significantly more successful than random guessing (the red dashed line).

**Setup and Notation.** We denote by $P$ the soft or discrete prompt that is prepended to any input sequence $x_i$ when querying the language model $L$. For brevity, we denote $L([P, x_i])$ by $L_P(x_i)$.[3] The output $y_i$ of $L_P(x_i)$ is an $M$-dimensional probability vector, with $M$ being the size of the model's vocabulary. Each component of $y_i$ corresponds to the probability that the $L_P$ assigns to the respective token for being the next token in the sequence $x_i$. The semantic meaning of the next token varies depending on the given downstream task. For instance, for classification, the index with the highest probability indicates the token of the class that $L_P$ assigns to $x_i$.

## 3 Private Information about Prompt Data Leaks from Prompted LLMs

By instantiating a MIA against prompted LLMs, we want to highlight that the private data used within a prompt (which we refer to as *prompt data* from hereon) can be subject to a substantial privacy risk. We showcase this risk at the example of LLMs that are prompted with discrete prompts $P$ containing tuples of demonstrations from classification downstream tasks as prompt data $p = \{(p_x, p_y)\}$. For example, in a prompt with one demonstration (*one-shot learning*), the prompt data $p$ may be specified as $p = \{("The movie was great.", "positive")\}$. Our prompts are provided in a consistent template where one or multiple demonstrations are combined with instructions as $P = [Instruction, (text sequence p_x, class-label token p_y), \ldots ]$.

For our MIA, we consider an adversary who aims at inferring whether a given private demonstration $(p_x, p_y)$ was used within the prompt data $p$. The adversary holds $n$ candidate demonstrations of text sequences and corresponding labels $l_i$ and queries the text sequences $(x_1, \cdots, x_n)$ to $L_P$ with black-box access. The prompted model $L_P$ then returns the output probability vectors $(y_1, \cdots, y_n)$. Following prior work [21, 53], we analyze the model's output probability at token $y_{i,l_i}$ that corresponds to the *correct* target class label of every $x_i$. The intuition to distinguish between members and non-members is that the output probabilities at the correct class $l_i$ will be significantly higher for demonstrations that were used within the prompt, *i.e.,* members with $(p_x, p_y) = (x_i, l_i)$. We show that even with this simple MIA, we can reliably determine membership for the prompt data.

**Experimental Setup.** We prompt GPT3-Babbage [6] with multiple *one-shot examples* to solve four standard downstream text classification tasks, namely *dbpedia* [57], *sst2* [46], *agnews* [57] and *trec* [50]. The template of our prompts follows [58]. To evaluate our MIAs, we consider the single data point used within the prompt as a members and 50 other randomly selected data points from the respective task's training dataset as non-members. This skewed distribution between members and non-members (1 vs 50) corresponds to a realistic scenario where only a small proportion of the candidate data targeted by the adversary are members [21]. To quantify the success of our attack, we report the AUC-ROC curves of 100 random trials.

---

[3]In the prefix method, $L_P$ denotes prepending trainable parameters to every layer's input, and not only to the model input $x_i$.

**Results.**  Before evaluating the success of the MIA, we analyze the probability output from GPT3 for the correct target class between member and non-member data points. Figure 2a shows for the dbpedia dataset that the prediction probabilities for non-members are significantly lower than for members. Figure 2b shows that this leads to a high MIA risk in terms of an average AUC score of 0.84 for the prompt data. Similar results for other datasets and models are presented in Appendix D. These results highlight that private information can leak from prompt data easily and thus motivate the urgent need for defenses which we develop in the rest of this paper.

## 4 Methods for Privacy Preserving Prompts

As of now, if we want to protect the private downstream data, we have to forego prompting altogether because, to the best of our knowledge, no algorithms for private prompt learning exist. The only alternative to privately adapt the LLM would be to perform DP fine-tuning [25, 54]. However, this approach is only feasible when we have direct access to the LLM to update its parameters with DPSGD [25] or to even change the model architecture to insert additional parameters—fine-tuned with DPSGD [54]. This is prohibitively expensive and mostly impossible with the commercial API, thus we propose the first algorithms that enable differentially private prompt learning.

We consider two main paradigms of prompting: soft prompts and discrete prompts. To learn private soft prompts, we introduce PromptDPSGD. PromptDPSGD is a parameter-efficient alternative to DP fine-tuning that does not need modifying the parameters or architectures of the LLM. However, many popular APIs [3, 6, 10, 17, 35] do not support soft prompts yet as it requires gradients with respect to the input. Therefore, we propose PromptPATE for discrete prompts. PromptPATE requires only black-box access to an LLM without any knowledge of the LLM's architecture or mode of operation. Instead, the algorithm only needs the next-token prediction of the LLM. This, to our knowledge represents the first solution for privately adapting LLMs in restricted API setups.

### 4.1 PromptDPSGD: DPSGD for Private Soft Prompt Learning

In general, all discrete input tokens to LLMs are internally transformed into continuous input embeddings that the LLM then operates on. Soft prompts are just additional continuous input embeddings that can be prepended to the original input embeddings before passing them through the LLM. To train (or *tune*) soft prompts, we require training data from a potentially private downstream task. After prepending the continuous soft prompt embeddings to input examples from the training data, we can calculate the gradients for the loss of the prompted LLM with respect to these soft prompt embeddings. The gradients provide information about how the soft prompt should be updated in order to minimize the loss on the training data.

If we can obtain the gradients for soft prompts, we can learn these prompts with privacy guarantees by applying the canonical DPSGD algorithm [1]. The same applies to prefix, therefore, when we talk about soft prompts in the following, we implicitly also include prefix. We call this approach PromptDPSGD. The algorithm yields soft prompts with DP guarantees that can be deployed with the LLM to solve the respective downstream task. The privacy analysis of PromptDPSGD follows the one of the standard DPSGD. Note, however, that while conceptually similar to fine-tuning the LLM's parameters with DPSGD [54, 25], PromptDPSGD differs in a crucial aspect. In DP-SGD fine-tuning, we require the gradients with respect to all or a subset of the model parameters and update these parameters to minimize the loss. In contrast, in PromptDPSGD, we use the gradients with respect to the soft prompt embeddings and only alter these. We highlight this difference in our PromptDPSGD-algorithm that we present in Appendix C.

While this difference seems subtle, it has far-reaching consequences. First, there are orders of magnitude fewer parameters that need to be updated which increases training efficiency. Second, and most importantly, it allows us to keep operating on the original LLM. We discuss the resulting advantages, such as storage efficiency, and the ability to process multiple different tasks simultaneously at the end of this section (in 4.3). These advantages make PromptDPSGD conceptually superior to private fine-tuning. At the same time, as we show in our evaluation, despite the small number of trainable parameters, PromptDPSGD, for simpler tasks, matches the performance of private fine-tuning. Yet, current APIs [3, 6, 10, 17, 35] do not support soft prompting, prefix, or private fine-tuning and only provide black-box access through discrete prompts. For these setups, we propose PromptPATE.

## 4.2 PromptPATE: PATE for Privacy Preserving Discrete Prompts

PATE [36, 37] enables learning classifiers with DP guarantees. It first trains an ensemble of *teacher* models on disjoint subsets of the private data. Second, through a noisy labeling process, the ensemble privately transfers its knowledge to an unlabeled public dataset. Finally, a separate *student* model is trained on this labeled public dataset for release. The noisy knowledge transfer in the second step relies on the Confident GNMAX algorithm [37] that we detail in Appendix C. It consists of three main parts: for any input data point from the public unlabeled dataset, each teacher votes for the most likely class. Then, the consensus over the teachers' votes is determined and queries with low consensus are rejected to avoid revealing too much information about the private decision boundary. Finally, the returned class label for any non-rejected data point is determined as a noisy argmax over all teachers' vote counts—where the added noise is sampled from a Gaussian distribution to implement the DP guarantees. For each rejected or labeled data point from the public dataset, privacy costs are accumulated and the ensemble stops labeling once a target privacy budget is reached.

Our PromptPATE follows the general flow of standard PATE: training the *teacher models*, *private knowledge transfer*, and training a *student model*. However, due to the significant differences between in-context learning for LLMs and supervised learning in the original PATE and how these different paradigms leverage private and public data, we had to redesign each of these building blocks. This allows to leverage both the data-efficiency of prompts and the rigorous privacy protection from PATE. In the following, we present the building blocks in our PromptPATE.

**Teacher Models (Flock of Stochastic Parrots).**   Instead of *training* teacher models on disjoint partitions of the private data, we use the private data to create disjoint prompts for the LLM. More specifically, we use examples, for instance {(*"The movie was great.", "positive"*), ...}, from the private training data to create prompts that can then be deployed with the LLM as teachers.

**Private Knowledge Transfer.**   During the private knowledge transfer, the teachers label public data sequences, such as (*"I did enjoy it."*, _). Each teacher votes with the most likely class labels for the private downstream task. In Appendix D, we show that PromptPATE can also operate directly on pure next token predictions from Claude [3] without access to per-token probabilities—enabling full black-box private prompts. By performing the private voting process according to standard PATE with the Confident GNMAX algorithm, we turn our per-teacher predictions into a final class label token that will be appended to the sequence, *e.g.,* (*"I did enjoy it"*, "positive"). The privacy accounting and analysis of our PromptPATE exactly follows the one of standard PATE [37].

**Student.**   The most naive way to obtain a student model following standard PATE would be to label many public sequences and train a language classifier using supervised learning on this data. However, due to the relatively high number of data needed for supervised learning, and the fact that each query to the private teachers consumes privacy, this process would incur high privacy costs. We propose a better approach building on the data-efficiency of prompting [42] by using labeled public sequences to create new discrete student prompts. The selected prompt can then be deployed with the LLM as the PromptPATE student model.

In theory, labeling one public sequence by the ensemble would be sufficient to create such a prompt. This approach yields negligible privacy costs, but the resulting prompt might not have good utility due to the high variance in the performance of prompts [58]. Therefore, we generate multiple prompts based on different labeled public sequences and perform prompt tuning to select the best student prompt. Care must be taken during selection: utility cannot be evaluated on the private data anymore given that the prompt will be publicly deployed and selecting based on the private data would incur additional privacy costs. We solve this tension by using parts of the newly-labeled public data as validation data to assess utility of the student prompts. By selecting the prompt with the highest validation accuracy, we deploy the student prompt that most resembles the private teachers.

## 4.3 Advantages of (Private) Prompting over (Private) Fine-Tuning

Our private prompt learning enables us to leverage the general advantages of prompting over fine-tuning while preserving privacy. Private prompting requires significantly less storage than private fine-tuning. While fine-tuning requires storing a separate copy of the LLM model for each downstream task [24], prompts operate only on the input level of LLMs without adapting model parameters, such that only a small task-specific prompt needs to be stored for each downstream task. For example, each copy of the fine-tuned RoBERTa base model requires 125M parameters (~500MB). This becomes

| Dataset | M | Soft-Prompt (Our) | | Prefix (Our) | | Full-Tuning [25] | | LoRA-Tuning [54] | |
|---|---|---|---|---|---|---|---|---|---|
| | P | <10K | | <100K | | 125M | | 1.2M | |
| | G | $\varepsilon = 8$ | $\varepsilon = \infty$ | $\varepsilon = 8$ | $\varepsilon = \infty$ | $\varepsilon = 8$ | $\varepsilon = \infty$ | $\varepsilon = 8$ | $\varepsilon = \infty$ |
| sst2 | | 92.31 | 95.64 | 91.97 | 96.33 | 85.89 | 96.40 | 92.97 | 96.60 |
| qnli | | 84.11 | 89.48 | 87.17 | 94.84 | 84.81 | 94.70 | 88.59 | 94.70 |
| qqp | | 81.52 | 86.56 | 82.58 | 91.42 | 86.15 | 92.20 | 86.26 | 92.20 |
| mnli | | 75.15 | 82.49 | 80.57 | 90.34 | 83.30 | 90.20 | 82.92 | 90.20 |

Table 1: **Performance of PromptDPSGD.** We report the accuracy values (%) for each dataset. All $\varepsilon$ values are reported as standard DP guarantees. We run the experiment on RoBERTa [30]. The first row **M:** the type of the private **M**ethod, the second row **P:** the number of **P**arameters tuned for the method, and the third row **G:** DP **G**uarantee. We also present results for $\varepsilon = 3$ in Appendix D.

prohibitively expensive, especially as the number of parameters for state-of-the-art LLMs rapidly increases. In contrast, soft-prompts and prefix, as the one generated by PromptDPSGD (using implementation from [29]) with the standard prompt length of 10 tokens require less than 10K parameters (40KB) for the soft-prompt and 100K parameters (400KB) for the prefix. A discrete prompt, such as the one generated in PromptPATE, requires less than 1 KB of prepended text. Prompts also enable processing many examples from different tasks in a single batch [26], called mixed-task inference. This allows more efficient use of LLMs since we do not have to wait for a sufficient number of requests for a single task before processing them. This is not possible with any form of fine-tuning, where the fine-tuned model can serve solely a single task.

## 5  Experimental Evaluation

We evaluate both PromptDPSGD and PromptPATE and show that they match the performance of non-private prompting while providing strong privacy guarantees.

### 5.1  PromptDPSGD

**Experimental Setup.**  To train soft-prompts and prefix, we follow the experimental setup from prior work on DP fine-tuning. Specifically, we use differentially-private optimization engines for transformers, such as models from the BERT family for the language understanding tasks. The experimental results for classification were performed on the RoBERTa models [30], using the standard NLP datasets, namely sst2, qnli, qqp, and mnli, from the GLUE benchmark [51]. Our implementation for soft-prompt and prefix is based on P-Tuning v2 [29]. To tune the (hyper)parameters for PromptDPSGD, we adjust the length of the soft-prompt or prefix in the private setting (with the default value of 10, which commonly yields good performance). For the privacy parameters, we set the $\delta = 1/N$, where $N$ is the number of data points in a given dataset, The clipping threshold of per-example gradients is set to 0.1 in most cases. We use a batch size of 1024. The detailed selection of (hyper-)parameters is presented in Appendix E.

**Results.**  We compare our PromptDPSGD against state-of-the-art approaches for private fine-tuning on multiple private downstream datasets. Our results are shown in Table 1. We highlight that both soft prompts and prefix provide competitive privacy utility trade-offs. For example, the difference in accuracy values between the non-private baseline and the private soft prompt ranges from 3% (for the simplest sst2 dataset) and up to 7% (for the most difficult mnli dataset). This mirrors results for other private methods, such as the private fine-tuning of LoRA [54]. We also observe that, similarly, for simple tasks, such as sst2 or qnli, the performance of soft prompt or prefix matches the one of fine-tuning. For the more difficult tasks, namely qqp and mnli, the performance of prefix and soft prompts is also relatively close to fine-tuning. The results obtained for these methods are highly influenced by the number of optimized parameters. For example, for the SST2 task and the RoBERTa-Base model, the prefix requires 19970 additional parameters while soft prompt adds solely 2306 parameters. On the other hand, the number of privately tuned parameters is a few orders of magnitude bigger for fine-tuning and equal to the size of the trained model, namely 125M for the method proposed in [25], while the fine-tuning approach from [54] optimizes around 1.2M

| Private | Lower Bound | Ens. Acc. | Upper Bound | Our PromptPATE | | | | | |
| | | | | **IID** Transfer | | | **OOD** Transfer | | |
| | $\varepsilon = 0$ | $\varepsilon = \infty$ | $\varepsilon = \infty$ | Public | $\varepsilon$ | Test acc | Public | $\varepsilon$ | Test acc |
|---|---|---|---|---|---|---|---|---|---|
| sst2 | 76.3 | 90.0 | 93.8 | sst2 | 0.178 | $88.8_{\pm2.3}$ | imdb | 0.187 | $87.2_{\pm1.9}$ |
| agnews | 62.0 | 72.8 | 78.2 | agnews | 0.248 | $71.7_{\pm0.8}$ | arisetv | 0.258 | $67.9_{\pm1.7}$ |
| trec | 40.7 | 57.6 | 58.7 | trec | 0.281 | $52.8_{\pm1.5}$ | qqp | 0.293 | $50.9_{\pm3.5}$ |
| dbpedia | 44.2 | 81.6 | 85.6 | dbpedia | 0.194 | $80.3_{\pm1.3}$ | agnews | 0.203 | $74.6_{\pm1.4}$ |
| sst2 (C) | 82.0 | 94.0 | 95.2 | sst2 | 0.147 | $92.3_{\pm1.1}$ | imdb | 0.154 | $92.7_{\pm0.8}$ |
| agnews (4) | 62.0 | 75.8 | 81.0 | agnews | 0.145 | $73.5_{\pm1.2}$ | arisetv | 0.145 | $69.6_{\pm1.8}$ |

Table 2: **Performance of PromptPATE.** We compare PromptPATE with three baselines: zero-shot (Lower Bound), the ensemble's accuracy (Ens. Acc), and the non-private baseline (Upper Bound) on four classification benchmarks. We study two settings, (IID Transfer) when the public dataset is from the same and (OOD Transfer) different distribution than the private data. We find that PromptPATE achieves strong privacy protection ($\varepsilon < 0.3$ at $\delta = 10^{-6}$) and utility close to the non-private and significantly higher than the zero-shot. Unless otherwise specified, the experiments are performed on GPT3-Babbage with one-shot prompts. Additionally, we also run experiments on GPT3-Curie for sst2 (C) and 4-shot prompts for agnews (4).

parameters. Our results reflect a general trend, where prompts are suited for small downstream tasks while fine-tuning with its bigger number of parameters can also cater to more complex tasks with larger training data sets.

## 5.2 PromptPATE

**Experimental Setup.** **Teachers:** Unless otherwise specified, we rely on GPT3-Babbage as the base LLM and select one-shot examples randomly without replacement from the private downstream task as prompt data. Our prompt template follows Zhao *et al.* [58]. For each setting, we deploy 200 teacher prompts. **Private knowledge transfer:** We use the implementation of PATE's Confident GNMAX algorithm and the privacy accounting from [12] and report our algorithm's hyperparameters in Appendix E. **Student:** For each private downstream task, we experiment with two setups (1) selecting public input sequences from the same (IID) and (2) from a different distribution (OOD) as the private data. We introduce three new datasets for the OOD setup: imdb [31], arisetv [11] and qqp [52]. The details of preprocessing these datasets can be found in Appendix E. In both the IID and OOD setup, we limit the size of the public dataset to 500 input sequences from the respective datasets. After the ensemble finishes labelling, we select the best labeled public sequence as prompt data based on the validation accuracy on the labeled public set. We repeat the process three times and report average and standard deviation of the test accuracy for the selected student prompt on the private test set. To improve utility, both teachers' and students' output probabilities from GPT3 are recalibrated using contexual calibration [58].

**Results.** We compare PromptPATE against three baselines: the lower bound baseline represented by a zero-shot prediction ($\varepsilon = 0$), *i.e.,* when the LLM is only prompted with an instruction, the private ensemble accuracy ($\varepsilon = \infty$), and the upper bound as a non-private one-shot prediction ($\varepsilon = \infty$) using the best example from the private data as prompt data. (To save costs, we select from 200 candidates.) Table 2 shows that, over all setups, PromptPATE achieves similar utility to the non-private baseline and significantly improves over zero-shot predictions—even at very strong privacy protection ($\varepsilon < 0.3$, $\delta = 10^{-6}$). Our results also highlight that the distribution of the public data does not need to be very close to the distribution of the private data to yield high-utility student prompts. For example, they can be collected from different domains (dbpedia holds extracts from wikipedia while its public data agnews contains news articles) and for different tasks (trec aims to classify the topic of a given answer while qqp serves to measure the similarity of two questions). Still, with dbpedia being the private downstream data and agnews as public, we achieve an accuracy of 74.6%, which is significantly higher than the zero-shot baseline with 44.2%.

We also provide further insights into the privacy-utility trade-offs that can be achieved with PromptPATE in Figure 3b. Our results highlight that with more public sequences queried to the ensemble, the privacy consumption increases while, after roughly 100 queries, with even $\varepsilon < 0.2$, the student model's test accuracy saturates. This yields very favorable privacy-utility trade-offs which we attribute

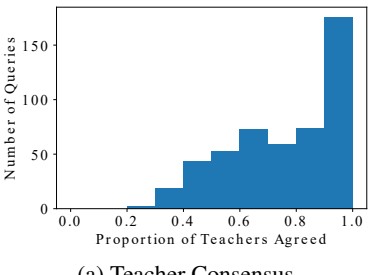

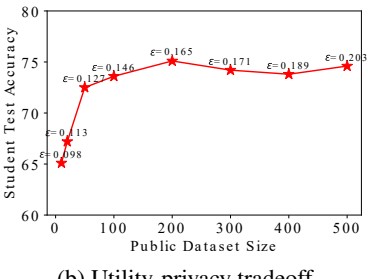

(a) Teacher Consensus

(b) Utility-privacy tradeoff

Figure 3: **Additional Insights of PromptPATE.** We perform ablation studies on GPT3-Babbage and use dbpedia as private and agnews as public data. *Left:* Teacher consensus as the fraction of teachers who vote for the correct class over 500 public input sequences. PromptPATE achieves overall high consensus. *Right:* Student accuracy as a function of the public query set's size. Already with as few as 100 queries, we observe a plateau in accuracy which highlights PromptPATE's data efficiency.

mainly to the data efficiency of discrete prompts: Even from within as little as 100 labeled examples, a high-performing student prompt can be derived. Additionally, we observe that the per-query privacy costs of PromptPATE are relatively low, further benefiting the privacy-utility trade-off. The small privacy costs result from the high consensus between the teacher predictions[4], see Figure 3a—that might result from all teachers relying on the same underlying LLM, just with different prompts.

**Scalability.** Finally, we also study how PromptPATE scales with larger LLMs and more examples in the prompt. We experiment with a more performant LLM (GPT3-Currie) for sst2. Due to the higher per-query costs, we are not able to repeat this experiment for all datasets. Our results show that the performance of our private prompt increases together with the performance of the public prompt (92.7% accuracy on Currie vs. 87.2% on Babbage) while the privacy budget $\epsilon$ decreases (from 0.178 to 0.147). To investigate flexibility in terms of numbers of private examples provided as prompt data, we also experiment for agnews with 4-shot teachers. Similar to the non-private study [58] that reports improvements for agnews in the 4-shot setting over 1-shot, we observe that this improvement also translates to the private prompt. Our results indicate that with increasingly more powerful LLMs and larger context windows, private prompting will increase further in terms of privacy-utility trade-offs.

## 6 Conclusions and Outlook

By instantiating the first simple yet effective membership inference attack against prompted LLMs, we show that they leak private information about their prompt data. We propose private prompt learning as a holistic and broadly applicable new approach to mitigate this risk. We first introduce PromptDPSGD that enables to train soft-prompts with privacy guarantees. In contrast to fine-tuning, soft prompts optimize significantly fewer parameters and do not require any update of LLM parameters or changes to its architecture. As the first solution to private downstream learning with LLMs in black-box access scenarios, we propose PromptPATE. PromptPATE builds on the highly data-efficient discrete prompts and implements privacy through a noisy knowledge transfer. Through our evaluation against two popular LLMs deployed behind commercial black-box APIs (GPT3 and Claude) [6, 3], we highlight that this method yields downstream performance that matches the one of non-private prompting at very strong privacy guarantees. As LLMs rapidly improve and increase in size, prompts are achieving consistently higher performance while fine-tuning becomes more challenging at this scale. This suggests that privacy protections for prompts will become even more important, especially as context sizes expand.

## Acknowledgments

We would like to acknowledge our sponsors, who support our research with financial and in-kind contributions: Amazon, Apple, CIFAR through the Canada CIFAR AI Chair, DARPA through the

---

[4]As motivated in Section 4.2, high consensus reveals less information about the private decision boundary, and hence incurs smaller privacy costs.

GARD project, Intel, Meta, NSERC through a Discovery Grant, the Ontario Early Researcher Award, and the Sloan Foundation. Resources used in preparing this research were provided, in part, by the Province of Ontario, the Government of Canada through CIFAR, and companies sponsoring the Vector Institute. We also thank members of the CleverHans Lab for their feedback.

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

# A   Broader Impacts

The growing importance of in-context learning as a paradigm for leveraging LLMs on private downstream tasks has significant implications for privacy. We present the first approaches for obtaining prompts with privacy guarantees, thereby enabling the use of this learning paradigm on sensitive data. This advancement has the potential to increase trust and acceptance of LLM-based systems for private applications. Our approach PromptPATE is the first viable technique for private downstream adaptation of black-box LLMs, which enables integrations into the state-of-the-art commercial LLM APIs. We acknowledge that—as with any application that relies on DP—care must be taken when choosing the privacy parameters $\varepsilon$ and $\delta$ since setting these incorrectly can lead to a false sense of privacy. Therefore, our work orientates at the privacy parameters that have been shown to provide reasonable protection in prior work. Thereby, we also ensure consistency and comparability in evaluations between the different appraoches.

# B   Limitations

**Tuning Instructions and Templates.**   For our discrete prompts, we did not tune the instructions or templates but instead relied on a template from prior work [58]. The effectiveness and performance of our PromptPATE could potentially be further improved by tuning the instructions and templates.

**Privacy Risk of Pretrained LLM:**   We build on pretrained LLMs to learn and deploy our private prompts. Our methods solely target the protection of the private data used for these prompts. However, it is also important to acknowledge the inherent privacy risks for data used to pretrain the LLM. We leave the pretrainig of LLMs with privacy guarantees to an orthogonal line of work.

**Limited Monetary Budget for our Experiments.**   Due to cost limitations, we were unable to experiment with the latest and best available model, GPT4. Our experiments with GPT3-Curie in comparison to less powerful GPT3-Babbage however indicate the clear trend the our private prompts improve in performance as the non-private baseline improves due to better models. Furthermore, again due to the cost limitation, we were not able to incorporate a larger number of teachers in our experiments for PromptPATE. Therefore, the best non-private teacher baseline that we report might not be the best achievable if one had more teachers to choose from. We chose from 200 and note that with more (and potentially better teachers), not only the baseline but also the teacher ensemble's performance would get better.

**Hyperparameter Tuning.**   To save computation costs, we did not exhaustively tune all hyperparameters in our experiments. While our approach still achieves high utility and good privacy-utility trade-offs, we acknowledge that with more hyperparameter tuning the performance together with the understanding of optimal configurations for private prompt learning could increase.

**Assumption of a Trusted LLM API Provider.**   In our work, the API provider gets to interact with the private data, for example, through the teachers' prompts in PromptPATE. Therefore, we have to assume trust in the API provider. The privacy guarantees through our private prompt learning protect the privacy of the prompt data against users that interact with the prompted LLM. In practice, companies that are concerned about the privacy of their data with respect to the API provider could make contracts with the API providers on the use of their data or buy access plans that guarantee that data queried to the API is treated privately. We leave implementing cryptographic approaches that could relief the assumption on trusting the API provider entirely, for example, by enabling the LLM to run inference on encrypted private data to future work.

# C   Additional Insights into our Methods

## C.1   PromptDPSGD

We present the full PromptDPSGD algorithm in Algorithm 1.

**Algorithm 1:** PromptDPSGD. In contrast to the standard DPSGD algorithm that updates model parameters during private training or fine-tuning, our PromptDPSGD privately updates the soft prompt parameters. We highlight these changes with respect to standard DPSGD training or fine-tuning in blue.

---

**Require:** Private downstream data $D = \{(x_i, y_i) \mid i \in [N]\}$, prompt sequence length $s$, embedding
   dimensionality $e$, trained LLM $L$ with frozen parameters, loss function $\ell(L_p, x)$ for prompted LLM,
   **Params:** learning rate $\eta_t$, noise scale $\sigma$, sampling rate $q$, max gradient norm $c$, training iterations $T$.
1: **Initialize** $P_0 \in \mathbb{R}^{s \times e}$ at random
2: **for** $t \in [T]$ **do**
3:   Sample mini-batch $B_t$ according to sampling rate $q$ from $D$ {Poisson sampling}
4:   For each $i \in |B_t|$, compute $\mathbf{g}_t(x_i) \leftarrow \nabla_{P_t} \ell(L_P, x_i)$ {Compute per sample gradient w.r.t. $p_t$}
5:   $\bar{\mathbf{g}}_t(x_i) \leftarrow \mathbf{g}_t(x_i) / \max\left(1, \frac{\|\mathbf{g}_t(x_i)\|_2}{c}\right)$ {Clip gradient}
6:   $\tilde{\mathbf{g}}_t \leftarrow \frac{1}{|B_t|}\left(\sum_i \bar{\mathbf{g}}_t(x_i) + \mathcal{N}\left(0, \sigma^2 c^2 \mathbf{I}\right)\right)$ {Add noise}
7:   $P_{t+1} \leftarrow P_t - \eta_t \tilde{\mathbf{g}}_t$ {Update soft prompt}
8: **end for**
9: **Output** $p_T$ and compute the overall privacy cost $(\varepsilon, \delta)$.

---

## C.2   PromptPATE

**Extended Background on PATE.**   We include the standard Confident-GNMax Aggregator Algorithm from [37] below.

---

**Algorithm 2:  Confident-GNMax Aggregator by [37]**

---

**Require:** input $x$, threshold $T$, noise parameters $\sigma_1$ and $\sigma_2$
1: **if** $\max_j \{\sum_{i \in [E]} n_{i,j}(x)\} + \mathcal{N}(0, \sigma_1^2) \geq T$ **then**
2:   **Output** $\arg\max_j \{\sum_{i \in [E]} n_{i,j}(\mathbf{x}) + \mathcal{N}(0, \sigma_2^2)\}$
3: **else**
4:   **Output** $\perp$
5: **end if**

---

## C.3   Privacy Analysis

**PromptDPSGD.**   Our PromptDPSGD can be seen as a repeated sampled Gaussian mechanism [1], with sampling performed over the entirety of the private prompt dataset. The difference to standard DPSGD for training or fine-tuning is that we do not update the model parameters, but the trainable embeddings for the soft prompts. This is conceptually different from standard DPSGD in terms of which parameters are updated. The privacy guarantees of the training mechanism still follow Abadi *et al.* [1], but with respect to the soft prompt embeddings: whether or not a particular data point will be included in the private training set used for tuning the prompt, the resulting soft prompt embeddings after training will be roughly the same. Especially by applying the clipping operation at every step, each mechanism's sensitivity is bounded by $c$. Privacy is then implemented as the trainable soft prompt embeddings are updated while adding noise noise drawn from $\mathcal{N}(0, c^2\sigma^2 I)$.

**Theorem 1** (Privacy of PromptDPSGD)**.** *Let $T$ be the total number of repetitions (training iterations) of our PromptDPSGD and the sampling rate be denoted by $q$. Then, there exist two constants $c_1$ and $c_2$, such that for any $\varepsilon < c_1 q^2 T$ our PromptDPSGD guarantees $(\varepsilon, \delta)$-DP, if for any $\delta > 0$, we choose the noise according to $\sigma \geq c_2 \frac{qc\sqrt{T \log 1/\delta}}{\varepsilon}$.*

*Proof.* The proof follows the one by Abadi *et al.* [1], using their moments accountant that models the privacy loss as a random variable dependent on the stochastic noise added. ☐

**PromptPATE.**   Our PromptPATE relies entirely on the Confident GNMAX algorithm from Papernot *et al.* [37]. We preserve the assumption underlying the algorithm and the respective privacy analysis that the sensitivity during the voting mechanism equals one. This is done in PromptPATE by assigning *disjoint* data points from the private prompt downstream dataset to all teachers. As a consequence, the privacy analysis of our PromptPATE entirely follows Papernot *et al.* [37].

Both our PromptDPSGD and PromptPATE experience the post-processing properties of DP, *i.e.,* once trained, the privacy guarantee $(\varepsilon, \delta)$ sets an upper bound on privacy leakage for the prompt data, independent on the number and type of queries that will be posed to the final prompted LLM.

# D  Additional Results

## D.1  Membership Inference Attacks

We present the full results of MIA against GPT3 with one-shot prompts on 4 datasets in SCW: Section4.

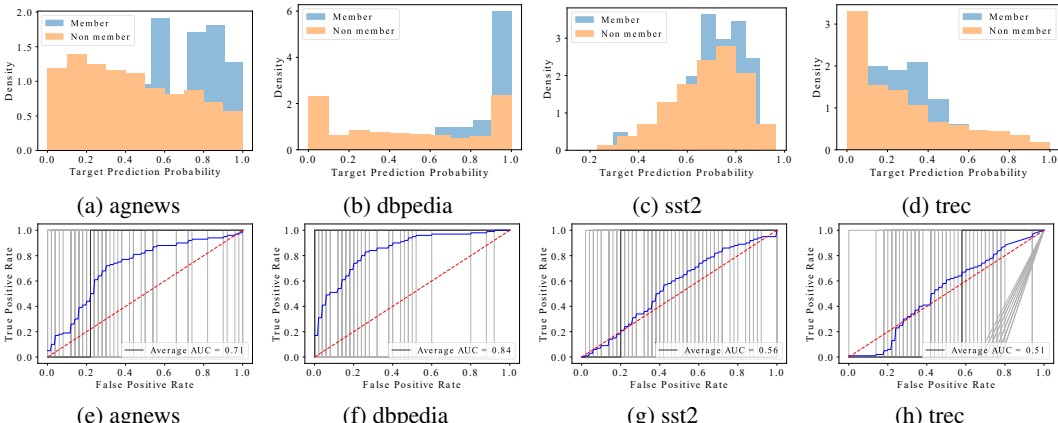

(a) agnews     (b) dbpedia     (c) sst2     (d) trec

(e) agnews     (f) dbpedia     (g) sst2     (h) trec

Figure 4: **MIA Risk over Multiple Datasets on GPT3.** We study GPT3-babbage prompted with 100 different one-shot examples on four datasets. *Top*: We present the prediction probabilities at the correct class for members (the one-shot example) and non-members (50 randomly sampled private points). The output probability for members is significantly higher than for non-member data points. *Bottom*: We present the AUC-ROC curves of our MIA against the 100 prompts (gray lines) and the blue line as an average over all attacks. Given that each prompt has only one member, the resulting TPRs can only be 0% or 100% which leads to the step-shape of the gray curves. The result indicates that our attack is significantly more successful than random guessing (the red dashed line).

In addition, we also perform similar experiments on GPT2-xl with four-shot examples, with results presented in Figure 5. We replace dbpedia with cb because the input in dbpedia is usually longer than the context length of GPT2.

## D.2  PromptPATE on Claude

We present the experiment results of PromptPATE on Claude [3]. Different from GPT3 that outputs logits over the whole vocabulary, Claude only gives us access to the next most likely token.

**Experimental Setup.**  **Teachers:** We rely on Claude-v1 as the base LLM. We use 2-shot prompts for sst2 and agnews, 4-shot for trec and 1-shot for dbpedia. We set the maximum generated tokens to 1 and temperatures to 0. We also create an "other" category in case the moel's output does not fall under any specified categories. For each setting, we deploy 400 teacher prompts. **Private knowledge transfer:** We use the implementation of PATE's Confident GNMAX algorithm and the privacy accounting from [12] and report our algorithm's hyperparameters in Appendix E. **Student:** We limit the size of the public dataset to 200 input sequences from the respective datasets. The number of shots for students corresponds with the teachers.

## D.3  More results for **PromptDPSGD**

We present the additional results for PromptDPSGD with $\varepsilon = 3$ on the classification tasks in Table 5.

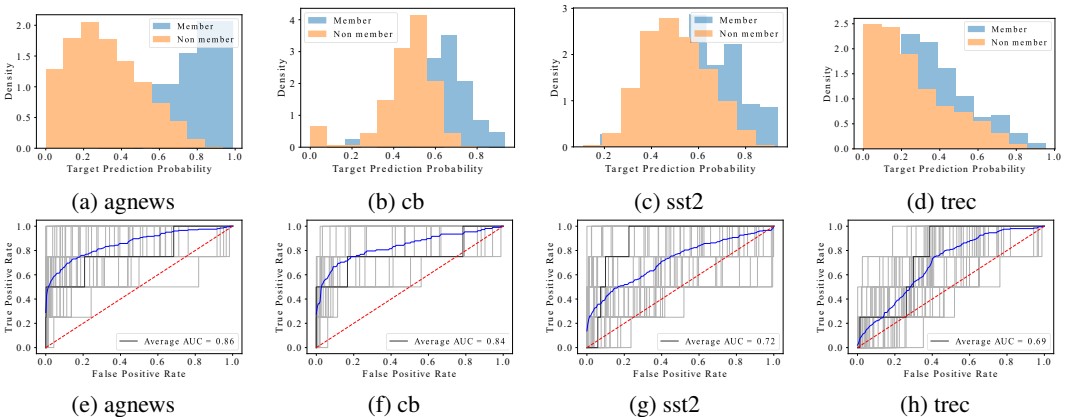

Figure 5: **MIA Risk over Multiple Datasets on GPT2-xl (4 shot).** We study GPT2-xl prompted with 100 different four-shot examples on four datasets. *top*: We present the prediction probabilities at the correct class for members (the one-shot example) and non-members (50 randomly sampled private points). The output probability for members is significantly higher than for non-member data points. *bottom*: We present the AUC-ROC curves of our MIA against the 100 prompts (gray lines) and the blue line as an average over all attacks. Given that each prompt has only one member, the resulting TPRs can only be 0%, 25%, 50%, 75% or 100% which leads to the step-shape of the gray curves. The result indicates that our attack is significantly more successful than random guessing (the red dashed line).

| | Lower Bound | Ens. Acc. | Upper Bound | Our PromptPATE | | |
|---|---|---|---|---|---|---|
| Private | $\varepsilon = 0$ | $\varepsilon = \infty$ | $\varepsilon = \infty$ | Public | $\varepsilon$ | Test acc |
| sst2 | 92.7 | 96.0 | 98.0 | sst2 | 0.048 | $95.7 \pm 1.4$ |
| agnews | 72.4 | 79.1 | 82.7 | agnews | 0.056 | $74.6 \pm 1.5$ |
| trec | 69.0 | 79.9 | 82.2 | trec | 0.068 | $79.3 \pm 1.2$ |
| dbpedia | 88.0 | 92.4 | 93.5 | dbpedia | 0.042 | $90.9 \pm 0.6$ |

Table 3: **Performance of PromptPATE on Claude.** We compare PromptPATE with three baselines: zero-shot (Lower Bound), the ensemble's accuracy (Ens. Acc), and the non-private baseline (Upper Bound) on four classification benchmarks. We find that PromptPATE achieves strong privacy protection ($\varepsilon < 0.1$ at $\delta = 10^{-6}$) and utility close to the non-private and significantly higher than the zero-shot.

# E   Additional Setup

## E.1   PromptDPSGD

We train PromptDPSGD on NVIDIA A100 GPUs. We execute (hyper-)parameter search that takes into account learning rate (LR), max grad norm (GRAD), number of epochs (Epochs), the token length of prefix and prompt. In general, we find that the prompt and prefix token length of 10 is close to the optimal value in most cases. For the private (hyper-)parameters, in most cases we tune for $\varepsilon = 8$ and use similar (or even the same) parameters for other $\varepsilon$ values. We set the max grad norm to 0.1 in most cases and then adjust the number of epochs (the more the better, for example, 100), and the learning rate [54][5]. The batch size is set by default to 1024.

We show the specific parameters chosen for PromptDPSGD in Table 6.

---

[5]We would like to thank the authors of [54] for their help, especially for the very useful and practical pieces of advice on how to tune the parameters for differential privacy from Huseyin A. Inan.

| Dataset | M P G | Soft-Prompt (Our) <10K | | Prefix (Our) <100K | | Full-Tuning [25] 125M | | LoRA-Tuning [54] 1.2M | |
| | | $\varepsilon = 3$ | $\varepsilon = \infty$ | $\varepsilon = 3$ | $\varepsilon = \infty$ | $\varepsilon = 3$ | $\varepsilon = \infty$ | $\varepsilon = 3$ | $\varepsilon = \infty$ |
|---|---|---|---|---|---|---|---|---|---|
| SST2 | | 90.48 | 95.64 | 90.37 | 96.33 | 91.86 | 96.40 | 92.60 | 96.60 |
| QNLI | | 83.62 | 89.48 | 86.05 | 94.84 | 87.42 | 94.70 | 86.97 | 94.70 |
| QQP | | 80.29 | 86.56 | 80.89 | 91.42 | 85.56 | 92.20 | 85.12 | 92.20 |
| MNLI | | 73.97 | 82.49 | 80.10 | 90.34 | 82.99 | 90.20 | 82.08 | 90.20 |

Table 4: **Private classification with soft prompts and prefix for** $\varepsilon = \{3, \infty\}$ **and the RoBERTa**$_{BASE}$ **model.** We use the same setup and notation as in Table 1.

| Dataset | M P | Soft-Prompt (Our) <10K | Prefix (Our) <100K | Full-Tuning [25] 125M |
|---|---|---|---|---|
| SST2 | | 91.05 | 93.58 | 90.94 |
| QNLI | | 87.62 | 89.45 | 89.42 |
| QQP | | 82.29 | 83.50 | 87.49 |
| MNLI | | 76.05 | 86.71 | 86.28 |

Table 5: **Private classification with soft prompts and prefix for** $\varepsilon = 8$ **and the RoBERTa**$_{LARGE}$ **model.** We use the same setup and notation as in Table 1.

## E.2 PromptPATE

### E.2.1 Hyperparameters for Confident-GNMax

We present our hyperparameters for Confident-GNMax in Table 7.

### E.2.2 Dataset Preprocessing

sst2, trec, agnews, dbpedia and cb are taken from the repo of [58]. All other public datasets are downloaded from huggingface. To reduce the cost of quering APIs, we randomly sample 300 points from the test set to report the test accuracy. For imdb, we random select one sentence from each entry and also remove the 
 tag. For qqp, we only take the column of "question 1" in the public set.

| Dataset | Method | RoBERTa | BS | LR | $\varepsilon$ | GRAD | Epochs | P-Length | Accuracy (%) |
|---|---|---|---|---|---|---|---|---|---|
| SST2 | Prompt | Base | 1024 | 0.005 | $\infty$ | N/A | 60 | 100 | 93.23 |
| SST2 | Prompt | Base | 900 | 0.05 | 8 | 0.01 | 21 | 9 | 92.32 |
| SST2 | Prompt | Base | 1024 | 0.005 | 3 | 0.05 | 100 | 10 | 86.35 |
| SST2 | Prompt | Large | 2048 | 0.005 | 8 | 4 | 100 | 10 | 91.05 |
| SST2 | Prefix | Base | 32 | 0.01 | $\infty$ | N/A | 60 | 20 | 94.61 |
| SST2 | Prefix | Base | 1000 | 0.05 | 8 | 4 | 22 | 1 | 91.97 |
| SST2 | Prefix | Base | 1024 | 0.01 | 3 | 0.2 | 100 | 50 | 90.37 |
| SST2 | Prefix | Large | 2048 | 0.05 | 8 | 4 | 22 | 1 | 93.58 |
| QNLI | Prompt | Base | 1024 | 0.005 | $\infty$ | N/A | 60 | 128 | 89.48 |
| QNLI | Prompt | Base | 1024 | 0.005 | 8 | 0.05 | 100 | 10 | 84.11 |
| QNLI | Prompt | Base | 1024 | 0.005 | 3 | 0.1 | 100 | 50 | 83.62 |
| QNLI | Prompt | Large | 2048 | 0.01 | 8 | 0.05 | 100 | 10 | 87.62 |
| QNLI | Prefix | Base | 1024 | 0.005 | $\infty$ | N/A | 60 | 20 | 94.84 |
| QNLI | Prefix | Base | 1000 | 0.03 | 8 | 0.07 | 22 | 10 | 88.77 |
| QNLI | Prefix | Base | 1024 | 0.01 | 3 | 0.2 | 100 | 50 | 85.78 |
| QNLI | Prefix | Large | 2048 | 0.03 | 8 | 0.07 | 22 | 10 | 89.45 |
| QQP | Prompt | Base | 1024 | 0.005 | $\infty$ | N/A | 60 | 50 | 86.64 |
| QQP | Prompt | Base | 1024 | 0.05 | 8 | 0.1 | 10 | 7 | 82.58 |
| QQP | Prompt | Base | 1024 | 0.001 | 3 | 0.01 | 100 | 15 | 80.29 |
| QQP | Prompt | Large | 2048 | 0.005 | 8 | 0.05 | 100 | 10 | 82.29 |
| QQP | Prefix | Base | 1024 | 0.005 | $\infty$ | N/A | 60 | 20 | 91.42 |
| QQP | Prefix | Base | 1024 | 0.05 | 8 | 0.1 | 10 | 7 | 82.59 |
| QQP | Prefix | Base | 1024 | 0.05 | 3 | 1 | 15 | 2 | 80.89 |
| QQP | Prefix | Large | 2048 | 0.05 | 8 | 0.1 | 10 | 7 | 83.50 |
| MNLI | Prompt | Base | 32 | 0.001 | $\infty$ | N/A | 60 | 20 | 82.49 |
| MNLI | Prompt | Base | 1024 | 0.005 | 8 | 0.05 | 60 | 10 | 75.01 |
| MNLI | Prompt | Base | 1024 | 0.005 | 3 | 0.05 | 100 | 10 | 73.97 |
| MNLI | Prompt | Large | 2048 | 0.005 | 8 | 0.2 | 60 | 10 | 76.05 |
| MNLI | Prefix | Base | 32 | 0.001 | $\infty$ | N/A | 60 | 20 | 82.49 |
| MNLI | Prefix | Base | 1024 | 0.005 | 8 | 0.05 | 60 | 50 | 80.42 |
| MNLI | Prefix | Base | 1024 | 0.005 | 3 | 0.2 | 100 | 50 | 80.10 |
| MNLI | Prefix | Large | 2048 | 0.01 | 8 | 0.1 | 100 | 10 | 86.71 |

Table 6: **Detailed parameters for soft prompts and prefix.** Type is the type of training, BS represents the batch size, LR denotes the learning rate, $\varepsilon$ is the DP guarantee, P-Length is the token length of soft-prompt or prefix.

| LLM | Dataset | $T$ | $\sigma_1$ | $\sigma_2$ |
|---|---|---|---|---|
| GPT3 | sst2 | 180 | 1 | 20 |
| GPT3 | agnews | 180 | 5 | 20 |
| GPT3 | trec | 180 | 1 | 20 |
| GPT3 | dbpedia | 170 | 1 | 20 |
| Claude | sst2 | 390 | 1 | 50 |
| Claude | agnews | 360 | 1 | 50 |
| Claude | trec | 320 | 1 | 50 |
| Claude | dbpedia | 320 | 5 | 50 |

Table 7: **Detailed parameters for Confident-GNMax.**

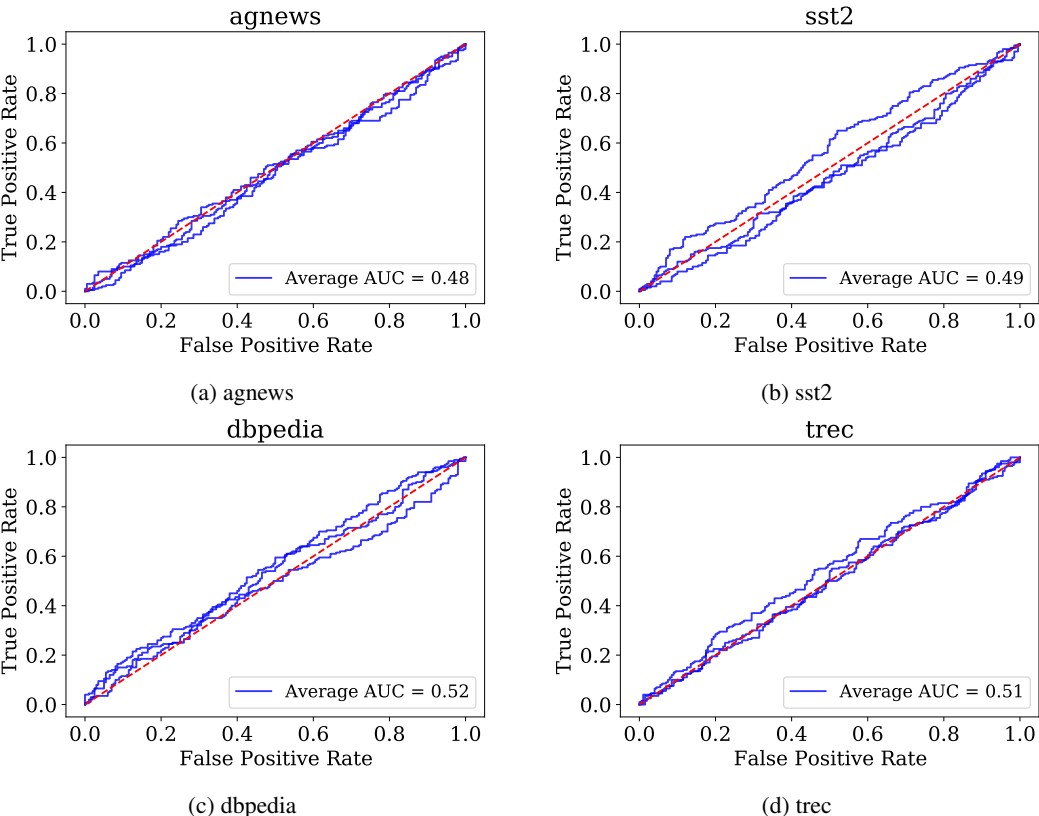

Figure 6: **MIA against the public prompts of PromptPATE.** We depict the AUC-ROC curve of MIA against the public prompts of PromptPATE. The member data is the examples from the prompts of all private teachers, and the non-members are randomly-selected data from the training set. Each blue curve corresponds to a different public prompt selected in one random trail. All curves are very close to the red dash line (random guess), which show that our PromptPATE is effective against MIA.

