# OpenReview forum: "Flocks of Stochastic Parrots: Differentially Private Prompt Learning for Large Language Models"
_NeurIPS.cc/2023/Conference — NeurIPS 2023 poster_

### Official Review · Reviewer_k8ve · 2023-06-30

**Soundness:** 3 good
**Presentation:** 3 good
**Contribution:** 2 fair
**Rating:** 6
**Confidence:** 4

**Summary:**

The paper studies the problem of differentially private prompting, i.e. the scenario where a prompt-augmented LLM is exposed to users, which should be able to interact with the model, but should not be capable of extracting the private prompt prepended to their queries. The authors first show, that a membership inference attack against example data used in a prompt is feasible and easy to carry out (given access to model logprobs). They then show how to adapt the DPSGD algorithm to enable differentially private soft prompt learning (PromptDPSGD) and devise a new method for differentially private prompt learning in the discrete setting (PromptPATE) based on teacher-student knowledge transfer. Their evaluation shows that both methods are effective in protecting prompt privacy without sacrificing too much model performance under reasonable \epsilon values.

**Strengths:**

* The paper is well-written and easy to follow, even though it covers a lot of ground. It motives the problem well by first demonstrating an effective MIA.

* The paper considers both the soft and discrete prompt learning setting. I think this is a good and pragmatic choice, as the most popular LLMs are API-guarded and typically do not provide access to gradients or hidden states, which are required for soft prompt learning.

* The paper explicitly considers efficiency and scalability of the proposed methods. This is important as state-of-the-art LLMs are very large and training them is expensive or even infeasible when accessible only via an API. Both the discrete and soft prompt learning methods are efficient and scalable, which is a major advantage over more data-hungry methods such as full model fine-tuning.


**Weaknesses:**

* The shown membership-inference-attack relies on model logits (log probabilities), which are not always available to users (or developers). This means the attack will not work with e.g. the latest OpenAI models like ChatGPT or GPT-4 [1]. Further, even with access to logits via API, model vendors often add noise to the logits to prevent distillation attacks.

* While stated later on, the paper initially does not clarify that PromptPATE is specifically designed for the relatively simple few-shot LLM classification setting. It is not clear how well the methods would work in other settings, e.g. when the model is used for free-form text generation. For PromptDPSGD this is less of an issue, as it seems to be more of a general method for prompt learning.

* I rated the contribution only as "fair", because after reading the paper it seems like PromptDPSGD is a rather simple adaptation of DPSGD and PromptPATE also is a relatively simple idea. It may be helpful if the authors could highlight the novelty of PromptDPSGD and PromptPATE compared with existing work.

* The paper does not cover the case where attackers want to extract instructional information from a prompt, e.g. when a prompt not only contains example data, but also a description of the task and the desired LLM output. From what I understand, PromptPATE would not work very well in this case, since even with teacher-student transfer, the resulting student prompt would still have to contain the same (likely human-readable) instructional information as the teacher prompt. I recognize this is as a different problem, but it may be worth mentioning in the paper, as common LLM use has shifted heavily from example-driven prompting to instruction-driven prompting, with the advent of instruction-tuned and RLHF models like ChatGPT.

[1] Chat API documentation, OpenAI, URL: https://platform.openai.com/docs/api-reference/chat/create

**Questions:**


* Can you comment on the effectiveness of your MIA attack against larger OpenAI models like `text-davinci-003`. From my own experience (which is anecdotal), logit noise is added to the API response for these larger models, which may make the attack less effective.

* Can you comment on MIA effectiveness with models that do not expose model logits at all, as is the case with ChatGPT and GPT-4 (see above)? Is prompt privacy still an issue in this case?

* Can you discuss the novelty of PromptDPSGD and PromptPATE compared to existing work like the standard DPSGD algorithm or existing discrete prompt optimisation approaches, to highlight the concrete contribution of the paper?

* Does PromptPATE and PromptDPSGD generalize to more free-form text generation tasks, e.g. when the model is used to generate longer text given a (private) prompt or exposed as a fully interactive chatbot? Can you protect against instruction extraction in these cases?

**Limitations:**

I think the paper should clarify early on, that it focuses on the restricted classification setting. This would help readers to better understand the specific problem the paper is trying to solve, where the more general prompt extraction problem should be considered as a separate problem.

---

> ### Author Rebuttal · Authors · 2023-08-09
>
> We thank the reviewer for their valuable feedback. Please find our detailed response in the following:
>
> >**1. The novelty of PromptDPSGD and PromptPATE compared with existing work.**
>
> - PromptDP-SGD: Our work is the first one to show the good utility of soft prompt tuning with DP-SGD. Compared with [25] and [54], our method optimizes orders of magnitude fewer parameters while keeping the original LLM frozen.  We leveraged libraries for DPSGD from full [25] and parameter-efficient [54] fine-tuning (LoRA) and combined them with the P-tuning V2 to be able to apply DP-SGD to the continuous prompts. Then, we carefully tuned the standard and privacy (hyper-)parameters, which resulted in good performance on many downstream tasks.
>
> - PromptPATE: This is the first method for DP learning with LLMs that requires only input-output access to the model. Also, our instantiation of each building block of PATE is novel and original. In the following, we list the novelty w.r.t each building block of PATE:
>
>   - Teachers: We are the first to observe how to leverage the effectiveness of in-context learning for the design of teachers. Instead of training teachers from scratch, we notice that the same LLM (with different prompts) can be instantiated as a teacher ensemble. This does not only make obtaining the teachers more efficient but also vastly decreases the required number of private data points.
>
>   - Student: The naive way of training a student for PATE would be to obtain many labels from the ensemble and then train a model in a supervised way. This would consume a large privacy budget due to a large public dataset needed for supervised learning. Therefore, instead of distilling teacher knowledge into a student model, we distill it into a single prompt, which significantly differs from the original paradigm of PATE. It enables us to obtain high-performance prompts with a small number of public labeled data points, making PromptPATE significantly better in terms of privacy-utility trade-offs than the naive adaptation of PATE.
>
>
> >**2. The effectiveness of MIA for the models that do not expose model logits at all,**
>
> Thank you for the comment. First, we would like to highlight that the main purpose of our MIA attack is not to present a powerful attack against modern LLMs but to serve as a motivational example that demonstrates private information from the prompts leaks through predictions of the prompted model. Our MIA shows that the predictions of the prompted LLM can be measurably influenced by the prompt data, which motivates the need for privacy-preserving prompt learning algorithms (the main focus of our paper). Previous works on label-only MIA (Choquette-Choo 2021, Li 2021) show that membership signals cannot be removed by publishing only labels. These attacks approximate the data’s distance to the decision boundary by making multiple queries on the perturbed inputs, which are shown to be as effective as the confidence scores. It is feasible to apply these member-only MIA to our prompt set-up. However, since it is not the focus of our paper, we’d like to leave it to future work.
>
> >**3. The effectiveness of MIA attack against larger OpenAI models that might add noise to the logits**
>
> Adding noises to the logits can decrease the success rate of MIA (in the limit, we can add enough noise to overshadow the membership signal) but this comes at the cost of lowering performance for legitimate users. Label-only MIA would be an effective attack against this defense strategy.
>
>
> >**4. Does PromptPATE and PromptDPSGD generalize to more free-form text generation tasks?**
>
> For PromptPATE, one way to extend it to generative tasks is by performing a private teacher vote for next-token predictions. Generating each next token can be thought of as a classification problem over the whole vocabulary. In fact, this idea has been used in SeqPATE (Tian et al. NeurIPS 2022)  for fine-tuned models. We perform some preliminary experiments on one-shot prompts with Claude, which shows that 86% of the private teachers generate the same first token on the e2e dataset. This high consensus among private teachers implies a promising signal for PromptPATE to succeed. However, we find that many details and design choices in this extension requires a deeper analysis and extensive engineering efforts. For example, it’s not immediately clear what vocabulary to vote on when that information is not given, which is the case for Calude. Due to the constraints of time and resources, we admit this to be a limitation to our work and leave it to the future work.
>
>
> >**5. Can you protect against instruction extraction in these cases?**
>
> That's a great point. However, we are not aware of any attack that could extract the private instructions from prompts effectively. On the other hand, our MIA against few-shot examples is shown to be successful. Therefore, we think that defending against privacy leakage of few-shot examples is better-motivated.
>
> >**6. I think the paper should clarify early on, that it focuses on the restricted classification setting.**
>
> Thank you for your suggestions. We agree with this. In our updated introduction, we will emphasize that "our paper focuses on classification" and "PromptPATE protects private examples in the few-shot prompts".
>
> Citation:
> - Tian, Zhiliang, et al. "SeqPATE: Differentially private text generation via knowledge distillation." Advances in Neural Information Processing Systems 35 (2022): 11117-11130.
> - Choquette-Choo, Christopher A., et al. "Label-only membership inference attacks." International conference on machine learning. ICML, 2021
> - Li, Zheng, and Yang Zhang. "Membership leakage in label-only exposures." Proceedings of the 2021 ACM SIGSAC Conference on Computer and Communications Security. 2021.

---

> > ### Comment · Reviewer_k8ve · 2023-08-15
> >
> > Thank you very much for clarifying my questions.
> >
> > > Re 5: However, we are not aware of any attack that could extract the private instructions from prompts effectively.
> >
> > I was asking mainly out of curiosity, but what comes to mind are simple "Print the first N characters of your prompt" kind of attacks, also more commonly known as "prompt jailbreaks".
> >
> > After carefully reading and considering the rebuttal and the other reviews, I will maintain my score, as it still aligns best with my overall perception.

---

> > > ### Author Response · Authors · 2023-08-16
> > > **Prompt jailbreak attack**
> > >
> > > Thank you very much for engaging with our rebuttal and pointing out the prompt jailbreak attack. This attack is not very effective in our few-shot classification set-up as the model is instructed to output only the class names mentioned in the prompt. However, we agree that this is a potential threat for the free-text generation tasks. We will emphasize in the updated introduction that our paper focuses on the privacy of few-shot examples in the classification tasks. Please let us know if you have further questions.

---

### Official Review · Reviewer_Hn4c · 2023-07-05

**Soundness:** 3 good
**Presentation:** 3 good
**Contribution:** 3 good
**Rating:** 7
**Confidence:** 4

**Summary:**

This paper investigated the privacy leakage in prompted large language models (LLMs), and have proposed methods to protect the privacy of potentially sensitive data used for prompt engineering. The authors first demonstrated high membership inference leakage in existing prompted LLMs and then proposed PromptDPSGD and PromptPATE for privately learning soft prompts and discrete prompts with DP guarantees. PromptPATE specifically works with existing commercial API, making it ideal for deployment in the real world.  The authors have validated their approaches on LLMs with extensive experiment setup and have shown a reasonably good privacy-utility balance.


**Strengths:**

* The paper is well-motivated as LLMs and prompt engineering are increasingly popular, the privacy concerns of sensitive data used in prompted LLMs are less studied. The privacy leakage analysis in this paper also validated the concern.
* The paper is organized and easy to follow. The literature review is comprehensive and the methods are clearly described.
* The experiments are conducted on LLMs with real world deployment scale and on black-box commercial API, showing the applicability of this research on existing real world LLMs systems. The setup of the experiment is extensive, including different scenarios such as private and public data from different domains.


**Weaknesses:**

Compared to the parameter-efficient fine-tuning approach (LoRA), prompt learning with DP has inferior performance.  Although the authors argued that the storage is cheaper with prompt learning, one might still prefer parameter-efficient LLMs based on their needs on the utility (also storage is typically considered cheap compared to other resources).


**Questions:**

* Why restrict PromptPate with much smaller epsilon < 0.2 while using larger epsilon (8) for PromptDPSGD? Do you not observe an increase in utility when increasing the privacy budget for PromptPate?
* Have you run membership inference on the DP protected models as a sanity check?


**Limitations:**

This paper has raised a privacy concern about real world deployment of LLMs. The authors have acknowledged limitations in the appendix of the paper.

---

> ### Author Rebuttal · Authors · 2023-08-08
>
> We thank the reviewer for their valuable feedback! Please find our detailed response in the following:
>
> >**1. One might still prefer parameter-efficient LLMs based on their needs on the utility.**
>
> We want to emphasize that the main advantage of prompt learning with DP is its flexibility and applicability even when the LLM is deployed behind an API, which is nowadays a standard (see for example the GPT family or Claude). DP with LoRA requires users to modify the model architectures, which almost all LLM APIs do not provide access to. In contrast, our PromptDPSGD only requires gradient access and PromptPATE only needs input-output access. These make our methods more practical and applicable without full access to LLMs.
>
> >**2. Why restrict PromptPate with much smaller epsilon < 0.2 while using larger epsilon (8) for PromptDPSGD?**
>
> Table 2 shows that our PromptPATE is very close to the accuracy of ensemble teachers, which is the upper bound of the student's performance. Figure 3.b further shows that the performance plateaus after $\epsilon=0.2$ and labeling more data points is not helpful. To further improve the utility of PromptPATE, it's crucial to improve the performance of the ensemble teachers, which is a separate question than privacy.
>
> >**3. Have you run membership inference on the DP protected models as a sanity check?**
>
> We thank the reviewer for this suggestions. During the rebuttal period, we ran our MIA on the public prompts over 3 random trials for each dataset. The average ROC scores over 3 trials are shown below. The AUC-ROC curves (each blue line corresponds with one student prompt for one trial) are in the pdf. The ROC scores are close to 0.5 and the curves are close to the random-guessing line, which show that PromptPATE effectively prevents the leakage from MIA.
>
>
> |     | sst2 | agnews | trec | dbpedia |
> |-----|------|--------|------|---------|
> | ROC | 0.49 ± 0.02 | 0.48 ± 0.03   | 0.51 ± 0.02 | 0.52 ± 0.04    |

---

### Official Review · Reviewer_96HV · 2023-07-07

**Soundness:** 4 excellent
**Presentation:** 4 excellent
**Contribution:** 3 good
**Rating:** 7
**Confidence:** 4

**Summary:**

This paper studies privacy preservation in the context of prompt learning for large language models (LLMs). It highlights the vulnerability of prompting data to membership inference attacks (MIAs) and proposes differential privacy (DP)-based defense methods for both soft and hard prompt learning. For soft prompts, the DP-SGD algorithm is utilized, while the PATE algorithm is adapted for hard prompts. Experiments on several common datasets and LM architectures demonstrate that the proposed algorithms can achieve good utility with a relatively small privacy budget.

**Strengths:**

S1. This paper presents a timely study on privacy preservation for prompt learning.

S2. The paper addresses both soft and hard prompt settings and proposes suitable differential privacy (DP) algorithms for each. The PromptPATE algorithm includes in-depth adaptations to the data-efficient characteristics of prompt learning.

S3. It presents MIA to motivate the DP-based defense algorithms.

S4. This paper provides systematic experiments on several common datasets and LM architectures to demonstrate the effectiveness of the proposed methods.

S5. Very well-written and easy to follow.

**Weaknesses:**

W1. PromptDPSGD is somewhat of a direct application of the original DPSGD to the soft prompt learning setting.



**Questions:**

I noticed that different language model architectures and datasets are used for PromptDPSGD and PromptPATE. Could you provide further discussion on this difference?

**Limitations:**

The authors have adequately discussed the limitations and potential negative societal impact of their work in the appendix.

---

> ### Author Rebuttal · Authors · 2023-08-08
>
> We thank the reviewer for their valuable feedback. Please find our detailed response in the following:
>
> >**1. PromptDPSGD is somewhat of a direct application of the original DPSGD to the soft prompt learning setting.**
>
> Our work is the first one to show the good utility of soft prompt tuning with DP-SGD. Compared with the [25] and [54], our method optimizes orders of magnitude fewer parameters while keeping the original LLM frozen.  We leveraged libraries for DPSGD from full [25] and parameter-efficient [54] fine-tuning (LoRA) and combined them with the P-tuning V2 to be able to apply DP-SGD to the continuous prompts. Then, we carefully tuned the standard and privacy (hyper-)parameters, which resulted in good performance on many downstream tasks.
>
>
> >**2. I noticed that different language model architectures and datasets are used for PromptDPSGD and PromptPATE.**
>
> The main reason why we use different models and datasets for soft and discrete prompts is to choose the most suitable set-up for each paradigm and keep consistent with each of its own previous works. As soft prompts can be applied to white-box LLMs, in order to provide a fair comparison with the previous DP fine-tuning methods [25,54], we use the same architecture (RoBERTa) and datasets as these two previous works. On the other hand, discrete prompts are mostly used with the decoder-only architecture like GPT3. In order to study the most suitable set-up for discrete prompts, our dataset choice and experiment design follows one of the pioneer works on in-context learning for GPT3 [58].
>
> Also, we want to emphasize that our paper does not aim to compare the performance of PromptDPSGD and PromptPATE. Instead, we want to provide alternatives for people to use in different scenarios. For example, one should use PromptDPSGD with the smaller and open-source LLM that offers gradient access and the downstream dataset size is large enough. On the other hand, PromptPATE should be used for larger and commercial LLMs (such as, GPT3 and Claude) has only the input-output access and the sensitive dataset is small.

---

> > ### Comment · Reviewer_96HV · 2023-08-12
> > **Thank you for the response.**
> >
> > I appreciate the author's rebuttal. I maintain my score and vote to accept.

---

> > > ### Author Response · Authors · 2023-08-12
> > > **Thanks for the feedback**
> > >
> > > Thanks for the positive feedback and voting to accept our paper. We deeply appreciate that.

---

### Official Review · Reviewer_RZeY · 2023-07-07

**Soundness:** 3 good
**Presentation:** 2 fair
**Contribution:** 3 good
**Rating:** 5
**Confidence:** 3

**Summary:**

This paper discusses the potential privacy risks associated with prompting data in large language models, which can be exposed through a membership inference attack. To address this issue, the authors propose two methods, PromptDPSGD and PromptPATE, for achieving private prompt learning. PromptDPSGD involves obtaining input gradients from the LLM and using FedAvg to update the soft prompt embedding while keeping the LLM frozen. On the other hand, PromptPATE creates a noisy ensemble of private discrete prompts and transfers knowledge by selecting a student prompt that can be publicly released. The experiments demonstrate that LLMs prompted with these private algorithms closely match the non-private baselines.

**Strengths:**

- The authors explore the potential privacy risks that may arise from prompting data in large language models, which can be exploited through a membership inference attack.
- The proposed approach for privately learning to prompt is novel and effective in preserving privacy while maintaining high accuracy.
- The experimental results demonstrate the effectiveness of the proposed approach in preserving privacy, which is important for real-world applications of large language models.


**Weaknesses:**

- The authors only verify the effectiveness of the proposed approach on NLU tasks. It is necessary to conduct experiments to verify the performance of the proposed methods on NLG tasks.
- I am not entirely convinced that prompting data poses significant privacy risks. In my view, the user input itself may present a more significant potential privacy risk.

**Questions:**

1. Could you provide more details of the membership inference attack? How to discover the member and non-member data points?
2. Would the proposed methods be applicable to NLG tasks? Additionally, have you tested the effectiveness of these methods on other NLU tasks?
3. I am not entirely convinced that prompting data poses significant privacy risks. From my perspective, the user input itself may present a more substantial potential privacy risk. What is your perspective on this? Are these methods also applicable to addressing privacy concerns related to user input?

---

> ### Author Rebuttal · Authors · 2023-08-08
>
> We thank the reviewer for their valuable feedback. Please find our detailed response in the following:
>
> >**1.  Could you provide more details of the membership inference attack?**
>
> We thank the reviewer for their question and are happy to provide more details on the membership inference attack. As we describe in Section 3, we are interested in the question if a given demonstration (private data point example) was used in a prompt.
>
> A prompt consists of a template that tells the model what to do, e.g. “Tell us if the following sentence is “positive” or “negative”, and example is “The movie was good”, “positive”. The data-label pair from the private data (“The movie was good”, “positive”) is the demonstration whose membership we are interested in. The question which data points *are* members, is therefore, determined by the chosen prompt. All other data points that are not used in the prompt are non-members.
>
> In standard MIA literature (e.g. [45]), the attacker who wants to find out which data points are the members  always has some candidate pairs of potential members. For example they would have the data points (“The movie was good”, “positive”), (“I liked the movie”, “positive”), (“What a terrible movie”, “negative”). Then, they would query all these candidates (only the sentences) to the model and observe the model’s confidence scores  on the labels (“positive” or “negative”) for the attack. Member data points have a higher confidence for the correct label.
>
> In our case, we assume that the attacker has a pool of potential member-candidates that consists of 1 member and 50 non-members. This skewed distribution is realistic given that the attacker has access to large amounts of data, out of which only a small fraction will be members of the concrete task. The attacker then queries x_1, … x_51 to the model and obtains the confidence scores z_1, … z_51  for the prompted model’s predictions at the right class belonging to x_i.
>
> Out of this vector of confidence values, we generate the ROC-curve (Figure 2b). We calculate the true positive rate (TPR) at different false positive rates (FPR). Given that only one of the 51 data points is a member point, there is a jump in the curve, where at a given threshold, the member data point is correctly classified as a member.
>
> We repeat this experiment over 100 different 1-shot prompts which yields 100 gray curves. We then average over these curves and plot the average as the blue curve. This then expresses the privacy risk of the prompted model for the private downstream dataset.
>
> >**2.  Would the proposed methods be applicable to NLG tasks?**
>
> In principle, PromptDPSGD can be directly applied to generative tasks. For PromptPATE, one way to extend it to generative tasks is by performing a private teacher vote for next-token predictions. Generating each next token can be thought of as a classification problem over the whole vocabulary. In fact, this idea has been used in SeqPATE (Tian et al. NeurIPS 2022)  for fine-tuned models. We perform some preliminary experiments on one-shot prompts with Claude, which shows that 86% of the private teachers generate the identical first token on the e2e dataset. This high consensus among private teachers implies a promising signal for PromptPATE to succeed. However, we also find that many details and design choices in this extension require a deeper analysis and extensive engineering efforts. For example, it is not immediately clear what vocabulary to vote on when that information is not given, which is the case for Calude. Due to the constraints of time and resources, we admit this to be a limitation to our work and leave it as our future work.
>
>
> >**3. I am not entirely convinced that prompting data poses significant privacy risks.**
>
> We would like to clarify that the threat-model for leakage of  private prompt data and private user inputs differs substantially. While the user input is leaked to the company who uses the prompted model and the API provider who deploys the LLM, the prompt data is already known to the company and leaks to the API provider, but **additionally**, when the company deploys the prompted model, it will leak to all other users of the prompted LLM. While we assume that the user shared their data with a company that they trust (or at least a company that has to follow the current privacy regulations) and while the API provider offers a service where the company can make contracts with them to not leverage the sent data for training with opt-out accounts (https://openai.com/blog/new-ways-to-manage-your-data-in-chatgpt), no guarantees can be given about other users (that are uncontrolled and potentially malicious). This significantly increases the risk for the private prompt data.
>
> Additionally, with pre-trained LLMs becoming more powerful, companies can leverage their users' private data more easily by deploying the model prompted with some examples of a task they want to solve. We identified privacy risk for data included in the prompts using the membership inference attack. Thus, the prompt provider can potentially expose private data via the textual prompts which is a severe privacy leakage.
>
> Citation:
> - Tian, Zhiliang, et al. "SeqPATE: Differentially private text generation via knowledge distillation." Advances in Neural Information Processing Systems 35 (2022): 11117-11130.

---

> > ### Comment · Reviewer_RZeY · 2023-08-18
> >
> > Thank you for the clarification. I genuinely appreciate the author's rebuttal, and it has influenced me to reconsider my review score (4->5). However, I still have some reservations regarding the potential privacy risks associated with prompting data, particularly in the case of LLMs utilizing SFT/RLHF methods like ChatGPT/GPT-4.

---

### Official Review · Reviewer_p7La · 2023-07-12

**Soundness:** 3 good
**Presentation:** 3 good
**Contribution:** 2 fair
**Rating:** 5
**Confidence:** 5

**Summary:**

This paper presents differentially private techniques for optimizing continuous and discrete prompts for solving text classification tasks using Large Language Models (LLMs). The need for differentially private prompt learning is motivated by showing that examples used in few shot prompts for classification tasks are highly susceptible to simple membership inference attacks

- For optimizing continuous prompts, the paper uses DP-SGD in combination with existing parameter efficient prompt tuning techniques. When used in combination with either soft prompt or prefix tuning, the method is generically called **PromptDPSGD** in the paper. This method requires access to gradients for tuning and the ability to adapt the model at inference time with tuned parameters, neither of which is typically supported in consumer APIs.

- For optimizing discrete prompts, the paper adapts the Private Aggregation of Teacher Ensembles (PATE) approach to label public examples using an ensemble of LLMs prompted with private few shot examples in order to learn a few shot prompt. This method is called **PromptPATE** in the paper. This method can be implemented on top of existing LLM completion APIs, even when they do not expose token prediction probabilities.

The paper evaluates the utility of using **PromptDPSGD** to fine-tune RoBERTa-Base on the SST-2, QNLI, QQP, and MNLI tasks from the GLUE benchmark with $\varepsilon \in$ {3,8}. It compares it to parameter efficient LoRA-tuning and full fine-tuning using DP-SGD. The results show that the accuracy of **PromptDPSGD** tuned models is competitive despite having orders of magnitude fewer tunable parameters than full fine-tuning and LoRA-tuning.

The paper evaluates **PromptPATE** using GPT-3 (Babbage and Curie) on SST-2, TREC, AG News, DBPedia. The results show that for $\varepsilon < 0.3$, it achieves similar accuracy to few-shot (1- and 4-shot) classification and to a non-private majority vote using the same teacher ensemble.

The supplemental material contains an Appendix describing hyperparameter choices, additional experimental results, and a discussion of broader implications and limitations. It also includes implementations of **PromptDPSGD** and **PromptPATE** built on existing libraries.

**Strengths:**

- Demonstrates for the first time that examples used in LLM prompts for text classification are susceptible to membership inference attacks.

- Evaluates the use of DP-SGD together with 3 different parameter efficient fine-tuning techniques for text classification tasks and compares it to full fine-tuning.

- Proposes a novel adaptation of PATE to optimize discrete prompts for solving text classification tasks using LLMs that is implementable using available commercial LLM APIs using a much lower privacy budget than fine-tuning techniques.

**Weaknesses:**

- Oversells the novelty of **PromptDPSGD**, which is just standard DP-SGD applied to existing parameter efficient fine-tuning techniques.

- The susceptibility of few shot learning to membership inference attacks has been studied before at least for image classification (https://openreview.net/forum?id=39kvovk9ju7). This diminishes the novelty of the observation that the same phenomenon holds for text classification.

- **PromptPATE** is applicable to classification tasks, but it does not appear generalizable to generative tasks where LLMs excel.

- **PromptPATE** performs worse than a non-private few-shot baseline in all scenarios evaluated. Table 2 shows that even a 4-shot **PromptPATE** has worse accuracy than the 1-shot non-private baseline. This casts doubt on the necessity of learning differentially private discrete prompts. It would be enough to declassify a single private prompt to get better utility with perfect privacy for the rest of the prompts.

- The choice to evaluate **PromptDPSGD** and **PromptPATE** on model classes with hugely different capabilities (RoBERTa, and GPT-3 and Claude, respectively), and on mostly disjoint tasks (with the exception of SST-2), makes it hard to compare the two approaches. Despite the difference in model capabilities likely making **PromptPATE** look better than **PromptDPSGD**, the results still show a large gap to a non-private baseline: e.g., for AG News, Table 2 reports 71.7 $\pm$ 0.8% with $\varepsilon = 0.248$ vs. 81% using a 4-shot prompt (which I think is a more appropriate baseline than a 1-shot prompt). The utility of **PromptPATE** cannot be significantly improved because it saturates at a lower privacy budget (cf. Figure 3b) compared to parameter efficient fine-tuning.

**Comments**

- In line 120, "To the best of our knowledge, no prior work attempted to provide DP guarantees for prompt data in LLMs." This peer-reviewed paper published shortly before NeurIPS 2023 deadline explores the use of differentially private prompt tuning in federated learning: https://doi.org/10.1109/ICASSP49357.2023.10095356.

- In line 322, The reference should be to Table 2 instead of Table 5.

- In line 539 in the Appendix: "trend the our private prompts" should read "trend that our private prompts".

- In Algorithm 1 in the Appendix: $D$ is specified first as a set of labeled examples $(x_i,y_i)$ but later only the features $x_i$ are used. In line 4, $L_P$ should read $L_{P_t}$ and $p_t$ should read $P_t$. In line 9, $p_T$ should read $P_T$.

- In Algorithm 2 in the Appendix: $\mathbf{x}$ should read $x$ and the parameter $E$ is not described.

- In the caption of Figure 5, "each prompt has only one member" should read "each prompt has only four members".

**Questions:**

a) Table 2 reports for AG News and IID public data 1-shot $\varepsilon = 0.248$ and 4-shot $\varepsilon = 0.145$. How can the privacy budget be lower for 4-shot than 1-shot?

b) I tried to wrap my head about how you produced the ROC curves in Figure 2b (and in the Appendix), but after reading the description in the paper many times I still do not fully understand it.

Do I get it right that your attack infers that $(x, l)$ is a member if and only if $\arg \max_j L_P(x)_j = l$?

Or does the attack take 51 different examples $(x_i, l_i)$ and infers that the example with index $\arg \max_i L_P(x_i)_{l\_i}$ is a member and the rest are not? Can you provide a pseudocode description of how the points in the ROC curve are computed?

**Limitations:**

Yes, both broader societal impact and limitations.

Appendix A discusses societal impacts. It highlights the risk of overreliance on DP guarantees and the need to correctly select the privacy hyperparameters $\varepsilon$ and $\delta$.

Appendix B discusses limitations. It highlights that privacy concerns are limited to private data in few-shot examples (as opposed to the training data of the LLM itself), the limited scale of experiments driven by cost constraints, and that private data used to construct prompts is exposed to the provider of the LLM API.

---

> ### Author Rebuttal · Authors · 2023-08-09
>
> We thank the reviewer for their valuable feedback. Please find our detailed response in the following:
>
> >**1. The susceptibility of few shot learning to membership inference attacks has been studied before at least for image classification**
>
> Thank you for pointing out this paper. We will add it to our related work section. However, we think our observations are novel since the few-shot learning algorithm studied in the paper is the traditional gradient-based fine-tuning, while ours is in-context learning. These two learning paradigms are vastly different. The main distinction is that in-context learning does not modify the model’s parameters whereas fine-tuning does. Given that fine-tuning modifies the model parameters according to the sensitive data, it is expected that the model then exposes membership information on that data. For our settings where the model's parameters are unchanged, the previous conclusion does not simply transfer.
>
> >**2.  It would be enough to declassify a single private prompt to get better utility with perfect privacy for the rest of the prompts.**
>
> We want to clarify that the 1-shot non-private baseline is with regards to the best 1-shot prompt selected based on the validation accuracy instead of a randomly-selected 1-shot prompt. [58] shows that the performance of in-context learning has huge variance, so a validation set is very important for utility. The privacy implications of this difference are also crucial and publishing a model with a 1-shot prompt cannot be seen as leaking only privacy of the chosen example. During the process of selecting the best 1-shot prompt, the model is prepended with multiple examples from the validation set, and the best performing one is chosen. As a consequence, the final prompt implicitly also contains private information on the non-chosen examples.
>
> >**3. Different models and datasets for PromptDPSGD and PromptPATE make it hard to compare.**
>
> The main reason why we use different models and datasets for soft and discrete prompts is to choose the most suitable set-up for each paradigm and keep consistent with each of its own previous works. Also, we want to emphasize that our paper does not aim to compare the performance of PromptDPSGD and PromptPATE. Instead, we want to provide alternatives for people to use in different scenarios.  Please see a more detailed response in the first point of the global response.
>
> >**4. Why is the privacy budget lower for 4-shot than 1-shot?**
>
> Our PromptPATE’s privacy analysis follows the standard PATE where the privacy budget depends on the number of teachers, the consensus among teachers and the size of the public set. The number of examples in the prompt does not directly influence the privacy budget. This is similar to standard PATE, where the privacy costs are not influenced by the private training set size
>
> >**5. How is the ROC curve produced?**
>
> We thank the reviewer for their question and are happy to clarify. We perform a confidence-based membership inference attack. The intuition is that for labeled samples (x,y), if (x,y) was used as a member of the prompt, when queried to predict on x, the prompted model will have a higher confidence on y than when (x,y) was not a member.
>
> In our case, we assume that the attacker has a pool of potential member-candidates that consists of 1 member and 50 non-members. This skewed distribution is realistic given that the attacker has access to large amounts of data, out of which only a small fraction will be members of the concrete task. The attacker then queries x_1, … x_51 to the model and obtains the confidence scores z_1, … z_51  for the prompted model’s predictions at the right class belonging to x_i.
>
> Out of this vector of confidence values, we generate the ROC-curve (Figure 2b). We calculate the true positive rate (TPR) at different false positive rates (FPR). Given that only one of the 51 data points is a member point, there is a jump in the curve, where at a given threshold, the member data point is correctly classified as a member.
>
> We repeat this experiment over 100 different 1-shot prompts which yields 100 gray curves. We then average over these curves and plot the average as the blue curve. This then expresses the privacy risk of the prompted model for the private downstream dataset.
>
>
> >**6. PromptPATE does not appear generalizable to generative tasks.**
>
> Thanks for your question. Please see the third point of our global response, where we explain how PromptPATE can be extended to generative tasks.
>
> >**7. 4-shot PromptPATE has worse accuracy than the 1-shot non-private baseline.**
>
> For agnews in Table 2, the improvement of 4-shot compared with 1-shot is not significant (< 3%). Thus, it’s not surprising that 4-shot private is worse than 1-shot non-private. Indeed, previous work [58] shows that having more examples in the prompts does not always increase the accuracy. For example, in Table 1 of [58], on the Trec dataset, 4-shot (69.7) gives worse accuracy than 1-shot (75.7). Therefore, we do not think it’s suitable to compare public and private setups with different number of shots.
>
>
> >**8. The utility of PromptPATE  saturates at a lower privacy budget compared to parameter efficient fine-tuning.**
>
> As shown in Table 2, PromptPATE is already very close to the teacher ensemble's performance, which is the upper bound of the student's performance. Therefore, if the teacher ensemble's performance can be improved, the utility of PromptPATE will improve as well.
>
>
> >**9. Related work of differentially private prompt tuning in federated learning**
>
> Thank you for pointing out this concurrent paper. We will add this work to the related work and change the sentence on line 120 “To the best of our knowledge, no prior work attempted to provide DP guarantees for **discrete** prompts in LLMs.”
>
> >**10 Comments 2-6**
>
> Thank you very much for pointing out the typos in the paper. We will update all of them in the new version of our paper.

---

### Author Rebuttal · Authors · 2023-08-08

We want to thank all reviewers for their feedback which has greatly helped us improve the paper. We are glad that the reviewers recognize our work to “present a timely study” (reviewer 96HV) on the private adaptation of LLMs “with real world deployment scale and on black-box commercial APIs” (reviewer Hn4c). Our proposed defense methods are “efficient and scalable” (reviewer k8ve), “novel and effective” (reviewer RZeY) with “a much lower privacy budget than fine-tuning techniques” (reviewer p7La). We hope that our work will contribute to a more trustworthy deployment of LLMs. Below we offer clarifications to some common questions reviewers have.

>**1. Why did you choose to compare PromptPATE and PromptDPSGD on different model architectures and datasets? (Reviewers: p7La, 96HV, Hn4c)**

The main reason why we use different models and datasets for soft and discrete prompts is to choose the most suitable set-up for each paradigm and keep consistent with each of its own previous works. As soft prompts can be applied to white-box LLMs, in order to provide a fair comparison with the previous DP fine-tuning methods [25,54], we use the same architecture (RoBERTa) and datasets as these two previous works. On the other hand, discrete prompts are mostly used with the decoder-only architecture like GPT3. In order to study the most suitable set-up for discrete prompts, our dataset choice and experiment design follows one of the pioneer works on in-context learning for GPT3 [58].

Also, we want to emphasize that our paper does not intend to compare the performance of PromptDPSGD and PromptPATE. Instead, we want to provide alternatives for people to use in different scenarios. For example, one should use PromptDPSGD with smaller and open-source LLMs that offers gradient access. On the other hand, PromptPATE should be used for larger and commercial LLMs (such as, GPT3 and Claude) that provide only input-output access and when the sensitive dataset is small.


>**2. Discuss the novelty of PromptDPSGD and PromptPATE compared to existing work (Reviewers: p7La, 96HV, and k8ve)**

- PromptDP-SGD: Our work is the first one to show the good utility of soft prompt tuning with DP-SGD. Compared with [25] and [54], our method optimizes orders of magnitude fewer parameters while keeping the original LLM frozen.  We leveraged libraries for DPSGD from full [25] and parameter-efficient [54] fine-tuning (LoRA) and combined them with the P-tuning V2 to be able to apply DP-SGD to the continuous prompts. Then, we carefully tuned the standard and privacy (hyper-)parameters, which resulted in good performance on many downstream tasks.

- PromptPATE: This is the first method for DP learning with LLMs that requires only input-output access to the model. Also, our instantiation of each building block of PATE is novel and original. In the following, we list the novelty w.r.t each building block of PATE:

    - Teachers: We are the first to observe how to leverage the effectiveness of in-context learning for the design of teachers. Instead of training teachers from scratch, we notice that the same LLM (with different prompts) can be instantiated as a teacher ensemble. This does not only make obtaining the teachers more efficient but also vastly decreases the required number of private data points.

   - Student: The naive way of training a student for PATE would be to obtain many labels from the ensemble and then train a model in a supervised way. This would consume a large privacy budget due to a large public dataset needed for supervised learning. Therefore, instead of distilling teacher knowledge into a student model, we distill it into a single prompt, which significantly differs from the original paradigm of PATE. It enables us to obtain high-performance prompts with a small number of public labeled data points, making PromptPATE significantly better in terms of privacy-utility trade-offs than the naive adaptation of PATE.

>**3. How do the presented methods PromptPATE and PromptDPSGD extend to other tasks, for example, to the text generation tasks? (Reviewers: p7La, RZeY, k8ve)**

In principle, PromptDPSGD can be directly applied to generative tasks. For PromptPATE, one way to extend it to generative tasks is by performing a private teacher vote for next-token predictions. Generating each next token can be thought of as a classification problem over the whole vocabulary. In fact, this idea has been used in SeqPATE (Tian et al. NeurIPS 2022)  for fine-tuned models. We perform some preliminary experiments on one-shot prompts with Claude, which shows that 86% of the private teachers generate the identical first token on the e2e dataset. This high consensus among private teachers implies a promising signal for PromptPATE to succeed. However, we also find that many details and design choices in this extension require more extensive analysis and engineering efforts. For example, it is not immediately clear what vocabulary to vote on when that information is not given, which is the case for Claude. Due to the constraints of time and resources, we admit this to be a limitation to our work and leave it as our future work.

We thank all reviewers again for their encouragements, feedback and comments. Please find the individual response to each reviewer below.

Citation:
- Tian, Zhiliang, et al. "SeqPATE: Differentially private text generation via knowledge distillation." Advances in Neural Information Processing Systems 35 (2022): 11117-11130.

---

### Decision · Program_Chairs · 2023-09-21

**Decision:**

Accept (poster)

**Comment:**

The paper contributes to the timely topic of privacy concerns in large language models (LLMs), particularly focusing on text classification tasks. It introduces novel variations of Differential Privacy via Stochastic Gradient Descent (DP-SGD) and Private Aggregation of Teacher Ensembles (PATE) to engineer privacy-preserving prompts. Despite some shortcomings, the paper is well-received by reviewers for its clear presentation, rigorous evaluation, and consideration of scalability and efficiency.

The paper presents one of the initial studies investigating the susceptibility of LLM-based text classification to membership inference attacks. The proposed PromptDPSGD and PromptPATE methods demonstrate a meaningful effort to bridge the gap between differential privacy and prompt engineering in the LLM setting. These methods' scalability and efficiency make them particularly relevant, given the computational constraints inherent in large-scale language models.

While the reviewers found the paper's contributions noteworthy, they also raised valid concerns about its limitations. The paper oversells the novelty in certain aspects, especially regarding PromptDPSGD, which is a somewhat incremental idea over existing DP-SGD techniques. I suggest this should be revised and contributions should be made more precise in the final version. Additionally, the paper mainly focuses on classification and leaves out other applications such as generation. This can also be clearly noted.

Overall, the paper provides a valuable contribution to the privacy challenges posed by LLMs. I believe it will make a nice addition to NeurIPS program. I encourage authors to refine their manuscript based on reviewer feedback for the camera-ready version.